# Ameliorative Effects of the Sesquiterpenoid Valerenic Acid on Oxidative Stress Induced in HepG2 Cells after Exposure to the Fungicide Benomyl

**DOI:** 10.3390/antiox10050746

**Published:** 2021-05-08

**Authors:** Mehtap Kara, Ezgi Öztaş, Tuğçe Boran, Ecem Fatma Karaman, Aristidis S. Veskoukis, Aristides M. Tsatsakis

**Affiliations:** 1Department of Pharmaceutical Toxicology, Faculty of Pharmacy, Istanbul University, Istanbul 34116, Turkey; ezgi.oztas@istanbul.edu.tr (E.Ö.); tugce.boran.1@istanbul.edu.tr (T.B.); ekaraman@biruni.edu.tr (E.F.K.); 2Department of Pharmaceutical Toxicology, Faculty of Pharmacy, Biruni University, Istanbul 34010, Turkey; 3Department of Nutrition and Dietetics, School of Physical Education, Sport Science and Dietetics, University of Thessaly, Argonafton 1, 42132 Trikala, Greece; veskoukis@uth.gr; 4Center of Toxicology Science and Research, Medical School, University of Crete, 71003 Heraklion, Greece; tsatsaka@uoc.gr

**Keywords:** valerenic acid, valerian, benomyl, oxidative stress, HepG2 cells, endoplasmic reticulum

## Abstract

Valerenic acid (VA) is a sesquiterpenoid and a phytoconstituent of the plant valerian used for sleeping disorders and anxiety. The frequency of using herbal components as therapeutic nutritional agents has increased lately. Their ability to improve redox homeostasis makes them a valuable approach against harmful xenobiotics. The purpose of this study was to evaluate the putative beneficial role of VA against the redox-perturbating role of the fungicide benomyl in HepG2 human liver cells in terms of oxidative stress in the cellular environment and in endoplasmic reticulum (ER). Benomyl increased cell total oxidant status and reactive oxygen species production and decreased total antioxidant status. The expression of genes coding for antioxidant molecules, namely, heme oxygenase-1, alpha glutathione s-transferase, NF-ĸB, and liver fatty acid binding protein, were decreased due to benomyl. VA ameliorated these effects. Benomyl also increased ER-stress-related molecules such as endoplasmic reticulum to nucleus signaling 1 protein, glucose-regulated protein 78, and caspase-12 levels, and VA acted also as a preventive agent. These results indicate that VA exerts ameliorative effects after benomyl-induced oxidative stress. VA, a widely used nutritional supplement, is a compound with potent antioxidant properties, which are valuable for the protection of cells against xenobiotic-induced oxidative damage.

## 1. Introduction

Valerian (*Valeriana officinalis*) is a plant native to Europe and Asia and has been used as a medicinal plant at least since the 5th century B.C. The *Valeriana* genus belongs to the family *Caprifoliaceae* that comprises approximately 200 species [1]. The extracts derived from the roots of the plant are commonly used for the treatment of sleeping disorders and anxiety since it has been shown that *V. officinalis* products have sedative and anxiolytic properties [2]. The German scientific advisory board of the Federal Institute for Drugs and Medical Devices (Commission E) has approved *V. officinalis* for the cure of sleeping disorders [3,4]. Valerian contains more than 150 chemical compounds with diverse physiological activities [2]. Valerenic acid (VA) is isolated from *V. officinalis*, and it has a sesquiterpene carboxylic acid skeletal structure, which is considered one of the major active compounds of the plant [4]**.** It has been reported that VA serum levels are associated with the pharmacologic effects of *V. officinalis* [5]. Noteworthily, several studies have demonstrated the antioxidant properties of extracts obtained from valerian species whereas VA is a food supplement commonly encountered in the market worldwide [6,7,8].

Apart from VA, the extracts that originate from valerian are rich in compounds, such as alkaloids, terpenes, organic acids, and flavones, with apparent nutritional value. Therefore, valerian extracts are also used as nutritional supplements [2]. This evidence implies that, probably, the preservation of redox homeostasis is a mechanism that allows VA, similarly to other medicinal plants, to act as an important nutritional (i.e., antioxidant) and pharmaceutical approach [9,10]. In this context, the use of natural products for health purposes due to their antioxidant properties has become widespread. Nevertheless, it is undoubtful that such substances could be proven to exert toxicity; therefore, their toxic potential should be evaluated very carefully [11,12]. Since herbal products can be used both alone or as part of a mixture, their beneficial and harmful profile assessment becomes more complex [10].

Benomyl is a fungicide that, although banned in developed countries, is still in use in developing countries. Belonging to the class of fungicides, it is a neurotoxic agent causing the degeneration of dopaminergic neurons [13,14]. A main toxicity mechanism of benomyl is the alteration in intracellular redox equilibrium inducing oxidative stress, acting similarly to other well-known pesticides and particularly chlorpyrifos and cypermethrin, as it has been demonstrated in several studies in mammals and other species [15,16,17]. In addition, benomyl stimulates oxidative stress also in cell lines by enhancing lipid peroxidation and by decreasing the reduced form of glutathione (GSH) [16]. Concerning the mechanisms by which benomyl could induce oxidative stress in the endoplasmic reticulum (ER), there are conflicting findings, a fact that confirms the largely unknown method of action of this fungicide at the molecular level [17,18,19,20,21]**.**

Based on the above, there is a gap in the literature regarding, on the one hand, the molecular mechanisms by which benomyl induces oxidative stress subcellularly and, on the other hand, the unknown method of action of VA. Therefore, the main objective of this study was to investigate the mechanism based on which benomyl induces its harmful redox-related effects and to examine the putative ameliorating effects of VA in HepG2 cells both in the whole cellular environment and at the ER level.

## 2. Materials and Methods

### 2.1. Chemicals

Benomyl (PESTANAL^®^, analytical standard (CAS Number: 17804-35-2) and valerenic acid (CAS No: 3569-10-6) were purchased from Merck (Germany). HepG2 cells were obtained from American Type Culture Collection (ATCC-HB8065, USA). DMSO and 3-(4,5-dimethylthiazol-2-yl)-2,5-diphenyltetrazolium bromide (MTT) were purchased from Sigma Chemical Co. Ltd. (USA). The H2DCFDA—Cellular ROS Assay Kit was purchased from Abcam (UK) (Cat No: ab133851). The chemicals for qPCR analysis were purchased from Roche Diagnostic (Germany). The SensiFast No-Rox Kit was obtained from Bioline (UK). The ELISA kits for human total oxidant status (Cat No: E1599Hu), human total antioxidant status (Cat No: E4350Hu), human DNA damage inducible transcript 3 protein (Cat No: E4838Hu), caspase 12 (Cat No: E0762Hu), human endoplasmic reticulum to nucleus signaling 1 (Cat No: E4839Hu), human glucose-regulated protein 78 (Cat No: E3624Hu), and human heat shock protein 90 kDa beta 1 (Cat No: E3012Hu) were all purchased from Bioassay Technology Laboratory (China). The glutathione (GSH) Elisa kit was purchased from Elabscience (USA) (Cat No: E-EL-0026). The cell culture mediums and all other supplements were purchased from Multicell Wisent (Quebec, Canada), and sterile plastic ware was purchased from Corning (Amsterdam, The Netherlands).

### 2.2. Cell Culture and Treatment Conditions

HepG2 cells (human liver cancer cell line) were maintained following the manufacturer’s instructions (ATCC-HB8065, USA). The cells were cultured in Eagle’s Minimum Essential Medium (EMEM) supplemented with 10% heat inactivated fetal bovine serum (FBS) and 1% antibiotic (100 U/mL penicillin and 100 µg/mL streptomycin) with 37 °C and 5% CO_2_ conditions. Stock solutions of benomyl and valerenic acid were prepared in 100% dimethyl sulfoxide (DMSO) and stored in −20 °C until the assays. During the experimental procedure, the stock solutions were diluted with cell culture medium to the desired concentrations, and the final DMSO concentration in the assay media was 1%. The HepG2 cells were treated with benomyl for 6 h, and then, they were treated with VA for 24 h in order to evaluate its putative effects on the response of the cell on benomyl-induced stress conditions.

### 2.3. Cell Viability Assay for Benomyl and Valerenic Acid

The concentrations of benomyl and VA used in this study were chosen based on the results obtained from the cell viability assay. The effects of benomyl and VA on cell viability were evaluated via the 3-(4,5-dimethylthiazol-2-yl)-2,5-diphenyltetrazolium bromide (MTT) test. The MTT tests were carried out separately for benomyl and VA. According to the process, 5 × 10^4^ cells in 100 µL of growth medium were incubated in 96-well plates for 24 h before treatment. Subsequently, benomyl and VA were administered to the cells separately. After exposure, the cells were incubated with MTT for 3 h, and after incubation, the medium was discarded and 100 uL of DMSO was added to each well. The absorbance was detected using a microplate reader (Biotek, Epoch, Vermont, USA) at 570 nm. For calculation of the IC_50_ and IC_30_ values of benomyl through the MTT assay, solutions with concentrations equal to 2.5, 5, 10, 20, 40, and 60 µM were used, whereas concerning VA, the concentrations of the tested solutions were equal to 300, 400, 500, 600, and 700 µM. After calculation of the IC_50_ values of benomyl and VA, further experiments were carried out to determine the IC_30_ values. The cells were exposed to 6 µM of benomyl (i.e., B6) for 6 h to achieve an induction of cellular stress. Then, in order to evaluate the effects of VA, 100 μM (VA100) and 200 µM (VA200) of VA were chosen for a 24 h cell exposure following benomyl exposure.

### 2.4. Total Oxidant Status Assay

The total oxidant status (TOS) assay was performed with the Human Total Oxidant Status Elisa Kit (Bioassay Technology Laboratory, China) following the manufacturer’s instructions. After exposure of the cells to benomyl and VA, they were detached using trypsin-Ethylenediaminetetraacetic (EDTA) and washed twice with ice-cold phosphate buffered saline (PBS). Then, the cells were centrifuged (1000× *g*, 20 min, 4 °C) and the supernatant was used for the TOS assay. The ELISA kit used included a 96-well plate that was precoated with human TOS antibody and a standard sample. Reconstituted standard solutions and sample supernatants were added into the 96-well plate at the desired volumes, and then, a biotinylated antibody was added only at the sample wells. Afterwards, streptavidin-horseradish peroxidase (HRP) was added into the wells and the samples were incubated at 37 °C for 1 h. Following, the wells were washed with washing buffer and the substrate A and B solutions were added so that the color development is proportionate to the TOS concentration. The reaction was terminated by the addition of a stop solution, and the absorbance was measured at 450 nm using a microplate reader (Biotek, Epoch, Vermont, USA). The TOS was calculated using a standard curve.

### 2.5. Total Antioxidant Status Assay

The total antioxidant status (TAS) assay was performed with the Human Total Antioxidant Status Elisa Kit (Bioassay Technology Laboratory, China) following the manufacturer’s instructions. After exposure of the cells to benomyl and VA, they were detached using trypsin-EDTA and washed twice with ice-cold phosphate buffered saline (PBS). Then, the cells were centrifuged (1000× *g*, 20 min, 4 °C) and the supernatant was used for the TAS assay. The ELISA kit used included a 96-well plate that was precoated with human TAS antibody and a standard sample. Reconstituted standard solutions and sample supernatants were added into the 96-well plate at the desired volumes, and then, a biotinylated antibody was added only at the sample wells. Afterwards, streptavidin-HRP was added into the wells and the samples were incubated at 37 °C for 1 h. Following, the wells were washed with washing buffer and the substrate A and B solutions were added so that the color development is proportionate to the TAS concentration. The reaction was terminated by the addition of a stop solution, and the absorbance was measured at 450 nm using a microplate reader (Biotek, Epoch, Vermont, USA). The TOS was calculated using a standard curve of solutions with known concentrations.

### 2.6. Total Reactive Oxygen Species Assay

The total reactive oxygen species (ROS) assay was performed with the abcam H2DCFDA—Cellular ROS Assay Kit (UK) following the manufacturer’s instructions. Based on the procedure, 25,000 cells per well were seeded in black 96-well plate applicable for a fluorescence microplate reader and allowed for 24 h to adhere. After incubation with benomyl and VA, the cell culture media was discarded and the cells were washed with 1 × kit buffer. Afterwards, 100 μL/well of diluted 2′,7′-Dichlorofluorescin diacetate (DCFDA) was added and the samples were incubated for 45 min at 37 °C in the dark. After incubation, the DCFDA solution was removed and 100 μL of 1 × kit buffer was added to each well and the fluorescence levels were measured at Ex/Em = 485/535 nm with a microplate reader spectrofluorometer (Biotek FLX-800). The results were expressed as a fold change from the assay control.

### 2.7. GSH Assay

GSH was measured through a competitive ELISA kit that included a 96-well plate that was precoated with human GSH antibody and a standard sample for the assay. After exposure of the cells to benomyl and VA, they were detached using trypsin-EDTA and washed twice with ice-cold phosphate buffered saline (PBS). Then, the cells were centrifuged (1000× *g*, 5 min, 4 °C) and the supernatant was used for the GSH assay. Reconstituted standard solutions and sample supernatants were added into the 96-well plate at the desired volumes and then a biotinylated antibody was added only at the sample wells. Afterwards, streptavidin-HRP was added into the wells and the samples were incubated at 37 °C for 45 min. Then, the wells were washed 3 times with the washing buffer, and afterwards, the HRP-conjugate was added and incubated for 30 min at 37 °C. Following incubation, the wells were washed 5 times and then the substrate reagent was added to all wells. After 15 min incubation at 37 °C, a stop solution was added and the absorbance was measured using a microplate reader (Biotek, Epoch, Vermont, USA) at 450 nm. The TOS was calculated using a standard curve of solutions with known concentrations.

### 2.8. Endoplasmic Reticulum Stress Biomarkers Assay

Human DNA damage inducible transcript 3 protein (DDIT3), caspase 12 (CASP12), human endoplasmic reticulum to nucleus signaling 1 (ERN1) protein, human glucose-regulated protein 78 (Grp78), and human heat shock protein 90 kDa beta 1 (Hsp90) as ER stress biomarkers were measured following the ELISA kit manufacturer’s instructions. After exposure of the cells to benomyl and VA, they were detached using trypsin-EDTA and washed twice with ice-cold phosphate buffered saline (PBS). Then, the cells were centrifuged (1000× *g*, 20 min, 4 °C) and the supernatant was used for the assays. All assays were performed separately. The ELISA kits included 96-well plates that was precoated with human biomarkers antibodies and standard samples for each assay. Reconstituted standard solutions and sample supernatants were added into the 96-well plate at the desired volumes, and then, a biotinylated antibody was added only at the sample wells. Afterwards, streptavidin-HRP was added into the wells and the samples were incubated in 37 °C for 1 h. Following, the wells were washed with washing buffer and the substrate A and B solutions were added so that the color development is proportionate to the DDIT3, CASP12, ERN1, Grp78, and Hsp90 concentrations. The reaction was terminated by the addition of a stop solution, and the absorbance was measured at 450 nm using a microplate reader (Biotek, Epoch, Vermont, USA). The TOS was calculated using a standard curve of solutions with known concentrations.

### 2.9. Gene Expression Analysis

After exposure of the cells with benomyl and VA, RNA isolation was performed with a High Pure RNA isolation Kit (Roche Diagnostic, Germany) according to the manufacturer’s instructions. A Transcriptor First Strand cDNA Synthesis kit (Roche Diagnostic, Germany) was used for cDNA synthesis. The gene expression levels of ERN1 (ID145212) and DDIT3 (ID 100355) were evaluated with a Real-Time Ready probe (ThermoFisher, UK) according to the manufacturer’s instructions, whereas nuclear factor kappa-light-chain-enhancer of activated B cells (NF-κB), heat shock protein 70 (Hsp70), liver-type fatty acid-binding protein (L-Fabp), heme oxygenase-1 (HO-1), and a-glutathione-S transferase (α-GST) were evaluated by custom-designed primers and SensiFast No-Rox Kit (Bioline, UK) on a qPCR platform (Light Cycler 480, Roche Diagnostic, Germany) (Table 1). Β-actin was used as a reference gene. The relative expression was calculated as previously described by Karaman and Ozden (2019) [22].

### 2.10. Statistical Analysis

Data were analyzed by one-way ANOVA and post hoc Dunnett’s t-test and expressed as mean ± standard deviation (SD). All analyses were performed using the statistical package SPSS version 20.0 for Windows (SPSS Inc., Chicago, IL, USA). The level of statistical significance was set at *p* ≤ 0.05.

## 3. Results

### 3.1. Cell Viability Assay

According to the MTT results, the IC50 values (i.e., the concentrations that reduced cell viability for 50%) for benomyl and VA were equal to 11.72 µM and 500.47 µM, respectively (Figure 1), whereas the IC30 values for benomyl and VA were equal to 6.4 µM and 392.36 µM, respectively. For calculation of the IC50 and IC30 values of benomyl, solutions of 2.5, 5, 10, 20, 40, and 60 µM were used, whilst regarding VA, solutions of 300, 400, 500, 600, and 700 µM were examined. Finally, the cells were exposed to 6 µM of benomyl for 6 h to achieve an induction of cellular stress. To evaluate the effects of VA, 100 and 200 µM of VA were chosen for 24 h cell exposure following the benomyl exposure period.

### 3.2. Total Oxidant Status (TOS)

The TOS levels were increased significantly in the B6-treated cells compared to the control cells. However, there were no significant difference in terms of TOS levels at B6+VA100 (6 μM of benomyl + 100 μM of VA) and B6+VA200 (6 μM of benomyl + 200 μM of VA)-treated cells compared to the control (Figure 2).

### 3.3. Total Antioxidant Status (TAS)

TAS levels was decreased significantly in the B6-treated cells compared to the control cells. However, there were no significant difference in terms of TOS levels at B6+VA100 (6 μM of benomyl + 100 μM of VA) and B6+VA200 (6 μM of benomyl + 200 μM of VA)-treated cells compared to the control (Figure 3).

### 3.4. Total Reactive Oxygen Species (ROS) Assay

Total ROS production was significantly increased in the B6-treated cells compared to the control cells (4.43-fold). Furthermore, ROS production was significantly decreased in B6+VA100 (6 μM of benomyl + 100 μM of VA) (2-fold) and B6+VA200 (6 μM of benomyl + 200 μM of VA) (3.8-fold)-treated cells compared to the B6-treated cells (Figure 4).

### 3.5. GSH Assay

The GSH levels were significantly decreased in the B6-treated cells compared to the control cells. Furthermore, the GSH levels were significantly increased in B6+VA100 (6 μM of benomyl + 100 μM of VA) and B6+VA200 (6 μM of benomyl + 200 μM of VA)-treated cells compared to the B6-treated cells. There were no significant differences between the control, and B6+VA100- and B6+VA200-treated cells (Figure 5).

### 3.6. Endoplasmic Reticulum Stress Biomarkers Assay

The levels of the biomarkers indicating ER stress, namely hsp90, DDIT3, Grp78, ERN1, and CASP12 were significantly increased in the B6-treated cells compared to control cells, whereas there were no significant differences between the control, and VA+100- and VA+200-treated cells (Figure 6).

### 3.7. Gene Expression Analysis

The HO-1 gene expression levels were significantly decreased in the B6-treated cells compared to control cells, whereas they were significantly increased in the B6 + VA200 (6 μM of benomyl + 100 μM of VA)-treated cells compared to control cells. L-Fabp gene expression levels were significantly decreased in the B6-treated cells compared to control cells and significantly increased in the B6+VA100 (6 μM of benomyl + 100 μM of VA) (2 fold) and B6+VA200 (6 μM of benomyl + 200 μM of VA)-treated cells compared to B6-treated cells. The α-GST gene expression levels were significantly increased in the B6+VA100 (6 μM of benomyl + 100 μM of VA) (2-fold) and B6+VA200 (6 μM of benomyl + 200 μM of VA)-treated cells compared to the control and the B6-treated cells (*p* < 0.05). The NF-κB gene expression levels were significantly decreased in the B6-treated cells compared to control cells, significantly increased in the B6-treated cells compared to control cells, and significantly increased in the B6+VA100 (6 μM of benomyl + 100 μM of VA) (2-fold) and B6+VA200 (6 μM of benomyl + 200 μM of VA)-treated cells compared to B6-treated cells. The ERN1 gene expression levels were dramatically increased in the B6-treated cells compared to all other cell samples; however, there were no significant difference between cell samples in terms of the DDIT3 and the hsp70 gene expression levels (Figure 7).

## 4. Discussion

Herbs and herbal extracts have been used for decades as traditional therapy due to their beneficial impact on cellular mechanisms [27]. Additionally, the use of herbal extracts as antioxidant products and as food additives and physical food conservatives have become more and more popular during the last few years [28]. It has been demonstrated that plant extracts and phytoconstituents exert their cell protective effects by scavenging free radicals or by enhancing antioxidant defense in vitro and in vivo [29,30,31,32,33,34,35]. Valerian is a plant native to Europe and Asia that has been used as a medicinal plant at least since the 5th century B.C. It is rich in a wide range of phytoconstituents, with amino acids, iridoids, alkaloids, flavonoids, and sesquiterpenoids being among them. VA is a sesquiterpenoid compound of valerian that has recently attracted the interest of research community since it is used as a compound in nutritional supplements. On the other hand, benomyl is a broad-spectrum fungicide that is mostly used in agriculture and in households for fungal growth inhibition. It is known that benomyl is a toxic agent for organisms, as it has been shown to induce neurotoxicity [16,36]. On that basis, the present study examined the potential protective role of VA on the harmful effects of benomyl using HepG2 cells as the experimental model. It is reported that benomyl induced ROS production and impaired the antioxidant defense of cells. The results were the same in the ER also. On the contrary, the concomitant treatment of the cells with two different VA concentrations resulted in amelioration of the negative (i.e., oxidative) effects of benomyl.

There are several studies in the literature that have revealed the sedative and anxiolytic effects of extracts derived from valerian root as well as phytoconstituents, such as VA, which has been mostly investigated for its beneficial role in the nervous system [4]. Indeed, it has been recently shown in in vitro and in vivo studies that it is a natural anxiolytic [37]. Furthermore, it appears that it acts protectively against glioblastoma cell growth and invasion, interestingly, through a redox-related molecular mechanism [38]. Additionally, it prevents neuroinflammation in Parkinson’s disease in mice, an important finding indicating an anti-inflammatory role of this natural component of *V. officinalis* [39]. It becomes evident that a potential molecular mechanism of VA action in vivo is through improving redox homeostasis, playing a putative role in NF-kΒ modulation. [40]. Its antioxidant effects, hence, have been revealed by a few studies; however, the molecular mechanisms have not yet been fully elucidated. [41,42].

Disruption of the intracellular redox equilibrium (i.e., induction of oxidative stress) is a phenomenon associated with onset of several diseases such as cardiovascular and neurodegenerative pathologies, endocrine system disruption-related diseases, and cancer [43,44]. Under stress conditions, nuclear factor erythroid 2-related factor 2 (Nrf2) leads to activation of the antioxidant response element, which regulates the expression of genes coding for antioxidant molecules, such as superoxide dismutase, glutathione peroxidase (GPx), glutathione S-transferase (GST), and heme oxygenase-1 (HO-1). HO and nuclear factor-kappa B (NF-κB) act as cell protectors against stress. Another cellular protection mechanism against stress is the activation of heat shock proteins (HSP) (e.g., hsp70 and hsp 90) [45,46,47,48]. During excessive cellular oxidative stress conditions, cells increase their antioxidant capacity via triggering the Nrf2/HO-1 pathway. Nrf2 plays an important role in regulating the mitogen-activated protein kinase (MAPK)-related inflammatory pathway by inhibiting NF-κB. This process results in downregulation of the inflammatory process, and additionally, Nrf2/HO-1 pathway activation induces an increase in anti-inflammatory cytokine action. Another important function of Nrf2 is regulating GSH synthesis enzymes such as GST, GPx, and glutathione reductase [48,49,50]. Natural compounds, such as nutrition constituents or nutraceuticals, have diverse effects on the Nrf2/KEAP-1/HO-1 pathway, although the mechanisms have not been described yet [51].

In this study, oxidative stress in HepG2 cells both systemically and in the ER was induced after exposure to benomyl and the ameliorative effects of VA, to which the cells were co-exposed with benomyl, were demonstrated. According to the results obtained, while ROS production and TOS levels were increased after exposure of the cells to 6 µM benomyl for 6 h, the GSH and TAS levels decreased. Benomyl also induced ER stress via the ERN1 pathway and caspase 12 induction. In addition, benomyl decreased the expression of genes coding for antioxidant molecules, namely as HO-1, L-Fabp, α-GST, and NF-κB. It must be noted that this is the first study to thoroughly investigate the toxic effects of benomyl at the molecular level in the cellular and ER environment. After 24 h exposure to benomyl alone, the cells were treated with benomyl plus VA at concentrations equal to 100 and 200 µM. According to the findings reported herein, it is shown for the first time that VA induces the expression of genes coding for antioxidant molecules, such as HO-1, L-Fabp, α-GST, and NF-κB in the tested cell line. Furthermore, VA decreases ERN1, grp78, and caspase-12 gene expression levels (Figure 8). The oxidative potential of benomyl in cellular and ER environments has been previously shown in human bronchial epithelial cells via IRE1, CHOP, and PERK induction, a fact that supports our results, although in a different cell model [17]. While there are studies suggesting the ameliorative effects of extracts originating from *V. officinalis* against cellular stress induction, the molecular mechanisms to which this biological action is attributed when measuring the expression levels of genes coding for key molecules have not been in depth examined [8,52,53,54]. It has been demonstrated that *V. officinalis*, and specifically VA, is a protective agent against rotenone-induced oxidative stress in neuroblastoma cells (SH-SY5Y cells), since it enhances the expression of genes coding for the antioxidant defense playing a crucial role in protection against Parkinson’s disease [42]. Another relevant study showed that VA helps a mouse model of Parkinson’s disease recover from a neuroinflammatory process [39]. It is evident that VA, similarly to other nutritional compounds, namely, whey protein, is an oxidative stress modulator with a beneficial role in human health [55,56,57]. The literature analysis indicates that there is scarce evidence about the potential ameliorating action of VA against the harmful, redox-related effects of benomyl; however, the molecular base of its biological action is largely unknown.

An important novelty of the present investigation is the examination of the antioxidant and compensatory role of VA against benomyl exposure on ER stress status. The ER is a crucial sub-cellular organelle that dynamically regulates normal protein synthesis and folding processes and, as a result, the biological action of cellular proteins. Under stress conditions, unfolded or misfolded proteins accumulate in the ER and trigger the activation of ER stress-related molecules to act protectively. Several different HSPs (HSP70, HSP90, and Grp78) play major roles regulating the response of ER stress. [58]. The following three major pathways knowingly participate in the ER stress process: inositol requiring enzyme 1 (IRE1-1), which in humans is encoded by the ERN1 gene (endoplasmic reticulum to nucleus signaling 1); protein kinase RNA-activated (PKR)-like ER kinase (PERK); and activating transcription factor 6 (ATF6). In normal conditions, Grp78 is inactive. During unfolded or misfolding protein response, grp78 initiates the PERK, IRE1-α, and ATF6 pathways. PERK activates the transcription factor CCAAT-enhancer-binding protein homologous protein (CHOP-DDIT3-gadd153) gene promoter region. If UPR is not eliminated, these molecules trigger the expression of proapoptotic proteins and caspase 12, resulting in apoptosis. Additionally, stressful conditions stimulate NF-κB to maintain cell survival. Noteworthily, ERN-1 is an NF-κB activator that provides connections between ER stress, oxidative stress, and inflammatory processes between the cell death process [59]. It has been reported that oxidative and ER stress interplay through Nrf2 activation is crucial for cellular events, such as apoptosis and cell survival [60].

## 5. Concluding Remarks

This study presents important findings concerning the ameliorating effects of VA in HepG2 cells after exposure to the fungicide benomyl. It is reported that benomyl induces oxidative stress both at the cellular and ER levels and that VA contributes to recovery of the cells by restoring their redox homeostasis. It appears that VA, a phytoconstituent of a medicinal plant (i.e., valerian), could be an effective phytochemical and a putative constituent of nutritional supplements with potent antioxidant and antitoxic properties, which are valuable in the protection of cells against the redox-related molecular damage induced by routinely encountered xenobiotics.

## Figures and Tables

**Figure 1 antioxidants-10-00746-f001:**
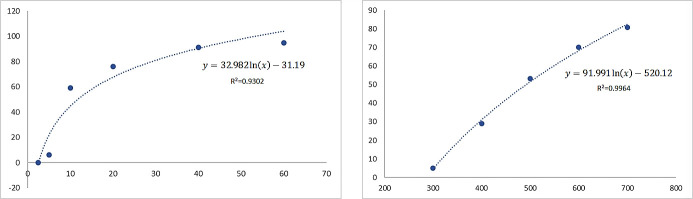
The effects of benomyl (**a**) and valerenic acid (VA) (**b**) on cell viability inhibition according to the 3-(4,5-dimethylthiazol-2-yl)-2,5-diphenyltetrazolium bromide (MTT) test results.

**Figure 2 antioxidants-10-00746-f002:**
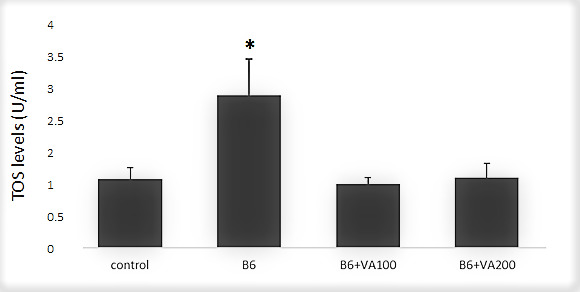
The effects of benomyl and benomyl accompanied by two concentrations of VA (100 and 200 μM) on the total oxidant status (TOS) levels of the cells. B6: exposure to 6 µM benomyl; VA100: exposure to 100 µM valerenic acid; VA200: exposure to 200 µM valerenic acid (* statistically significant difference compared to the control).

**Figure 3 antioxidants-10-00746-f003:**
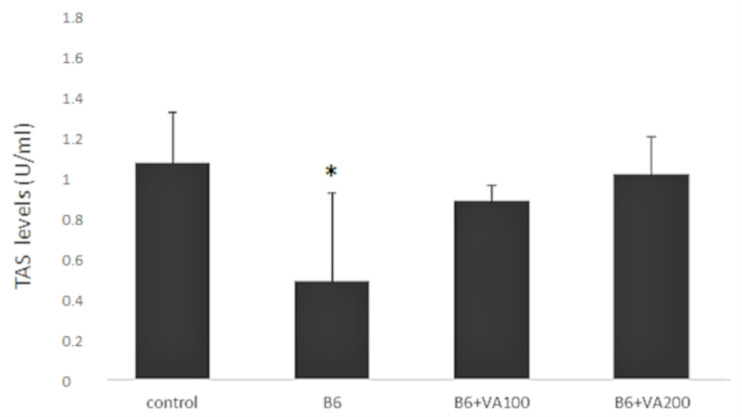
The effects of benomyl and benomyl accompanied by two concentrations of VA (100 and 200 μM) on the total antioxidant status (TAS) levels of the cells. B6: exposure to 6 µM benomyl; VA100: exposure to 100 µM valerenic acid; VA200: exposure to 200 µM valerenic acid (* statistically significant difference compared to the control).

**Figure 4 antioxidants-10-00746-f004:**
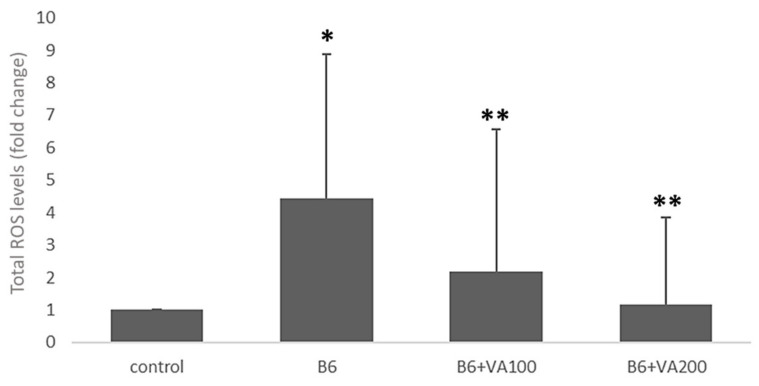
The effects of benomyl and benomyl accompanied by two concentrations of VA (100 and 200 μM) on the total reactive oxygen species (ROS) levels of the cells. B6: exposure to 6 µM benomyl; VA100: exposure to 100 µM valerenic acid; VA200: exposure to 200 µM valerenic acid (* statistically significant difference compared to the control; ** statistically significant difference compared to the B6 group).

**Figure 5 antioxidants-10-00746-f005:**
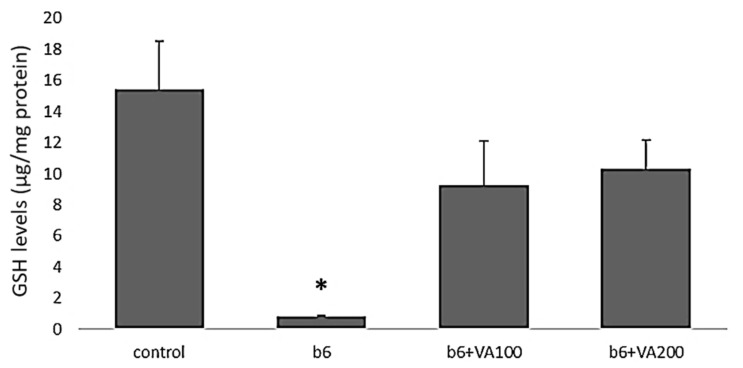
The effects of benomyl and benomyl accompanied by two concentrations of VA (100 and 200 μM) on the GSH levels of the cells. B6: exposure to 6 µM benomyl; VA100: exposure to 100 µM valerenic acid; VA200: exposure to 200 µM valerenic acid (* statistically significant difference compared to the control).

**Figure 6 antioxidants-10-00746-f006:**
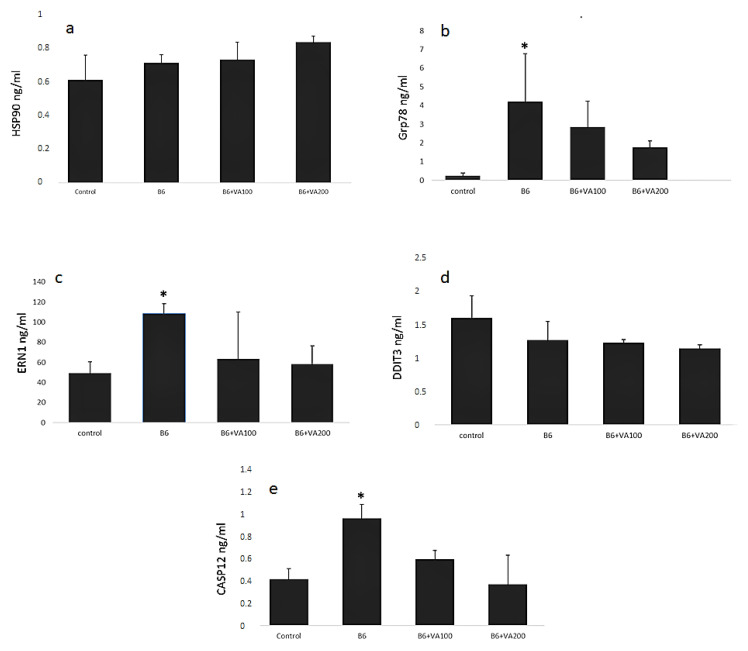
The effects of benomyl and benomyl accompanied by two concentrations of VA (100 and 200 μM) on the levels of molecules indicating endoplasmic reticulum stress of the cells. (**a**) human heat shock protein 90 kDa beta 1 (Hsp90) levels did not significantly changed; (**b**) human glucose-regulated protein 78 (Grp78) levels. Grp78 levels significantly increased in B6 group, however Grp78 levels in VA+100 and VA+200 groups nonsignificantly decreased; (**c**) human endoplasmic reticulum to nucleus signaling 1 (ERN1) levels significantly increased in B6 group, however ERN1 levels in VA+100 and VA+200 groups nonsignificantly decreased. (**d**) damage inducible transcript 3 protein (DDIT3) levels did not significantly changed; (**e**) caspase 12 (CASP 12) levels significantly increased in B6 group, however CASP12 levels in VA+100 and VA+200 groups nonsignificantly decreased. B6: exposure to 6 µM benomyl; VA100: exposure to 100 µM valerenic acid; VA200: exposure to 200 µM valerenic acid (* statistically significant difference compared to the control).

**Figure 7 antioxidants-10-00746-f007:**
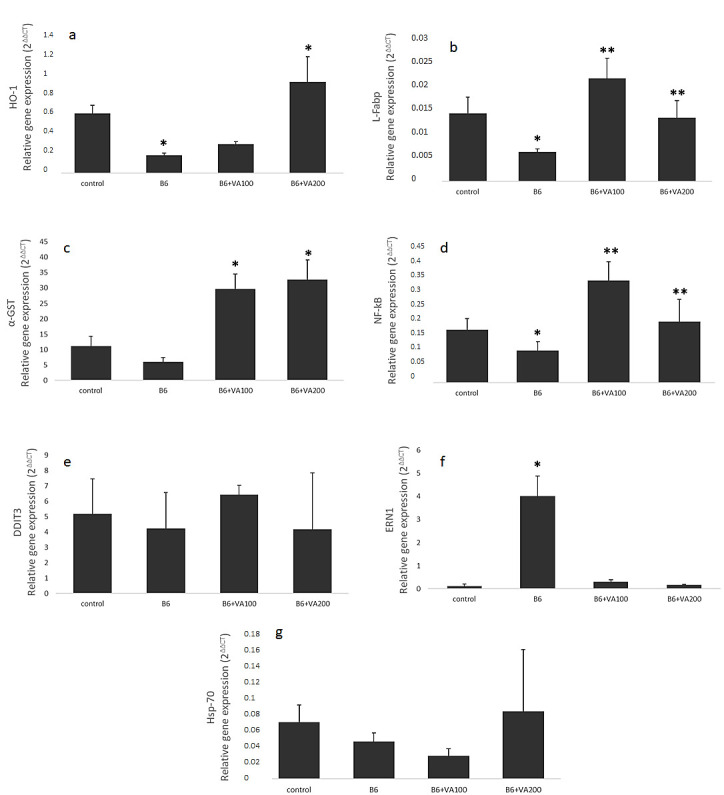
The effects of benomyl and benomyl accompanied by two concentrations of VA (100 and 200 μM) on the relative gene expression levels of oxidative stress and endoplasmic reticulum stress-related molecules of the cells; (**a**) HO-1: heme oxgenase-1; (**b**) L-Fabp: liver-type fatty acid-binding protein; (**c**) a-GST: a-glutathione-S transferase; (**d**) NF-κB: nuclear factor kappa-light-chain-enhancer of activated B cells; (**e**) DDIT3: human DNA damage inducible transcript 3 protein; (**f**) ERN1: human endoplasmic reticulum to nucleus signaling 1; (**g**) Hsp70: heat shock protein 70. B6: exposure to 6 µM benomyl; VA100: exposure to 100 µM valerenic acid; VA200: exposure to 200 µM valerenic acid (* statistically significant difference compared to the control; ** statistically significant difference compared to the B6 group).

**Figure 8 antioxidants-10-00746-f008:**
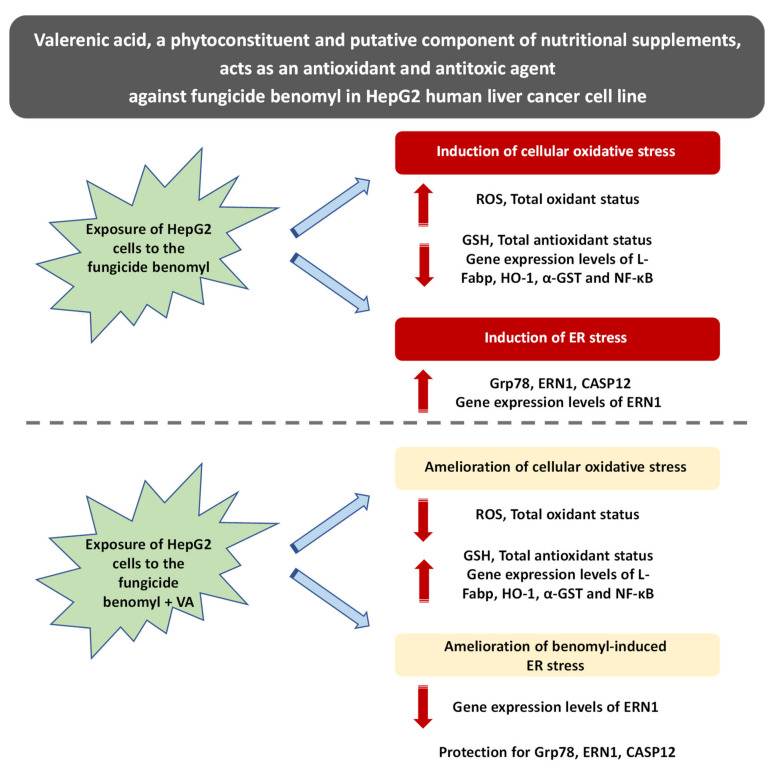
A graphical illustration of the main findings reported in the present study, according to which VA, a phytoconstituent that exerts ameliorative effects against oxidative stress induced by the fungicide benomyl at the cellular and ER levels in HepG2 cells. VA: valerenic acid; ER: endoplasmic reticulum; GSH: reduced form of glutathione; ROS: reactive oxygen species; L-Fabp: liver-type fatty acid-binding protein; HO-1: heme oxgenase-1; a-GST: a-glutathione-S transferase; NF-κB: nu-clear factor kappa-light-chain-enhancer of activated B cells; Grp78: human glucose-regulated protein 78; ERN1: human endoplasmic reticulum to nucleus signaling 1; CASP12: caspase 12.

**Table 1 antioxidants-10-00746-t001:** Primer sequences and annealing temperatures.

Gene.	Sequence (Forward 5′–3′)	Sequence (Reverse 5′–3′)	Tm	Ref
Hsp70	5′-CCGGTTCCCTGCTCTCTGTC-3′	5′-CAGTCCACTACCTTTTTCGAGA-3′	60	[23]
HO-1	5′-ATGACACCAAGGACCAGAGC -3′	5′-GTGTAAGGACCCATCGGAGA-3′	60	[24]
α-GST	3′-TTCCTTGGGCTGCCAGGC-5′	3′-GTTCCAGCAAGTGCCATCC-5′	60	[25]
NF-κB	5′-CATTGCTCAGGTCCACTGTC-3′	5′-CTGTCACTATCCCGGAGTTCA-3′	60	[26]
L-Fabp	5′- TGTCGGAAATCGTGCAG-3′	5′- GATTATGTCGCCGTTGAGTT-3′	53	[22]
β-Actin	5′- AACTACCTTCAACTCCAT-3′	5′- TGATCTTGATCTTCATTGTG-3′	48	[22]

## Data Availability

All data are comprised in this article.

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
