# Peer review of "Ameliorative Effects of the Sesquiterpenoid Valerenic Acid on Oxidative Stress Induced in HepG2 Cells after Exposure to the Fungicide Benomyl"

_antioxidants, 2021, doi:10.3390/antiox10050746_

Round 1

Reviewer 1 Report

In this interesting article, the authors investigated in vitro the oxidative stress-related effects of the fungicide benomyl and the possible protective effect of valerenic acid.

The research described in this manuscript has been correctly planned and well described.

I have only few minor points:

- Were the doses used of VA and benomyl chosen on the basis of previous research?

Please write a sentence justifying the doses tested.

- The discussion is too long. It should be more concise to make it easier to read.

- In the discussion I suggest adding a summary scheme of the results obtained.

Author Response

Dear editor,

We would like to thank you for your response and for giving us the opportunity to improve and re-submit the paper (antioxidants-1166645) entitled "Ameliorative Effects of the Sesquiterpenoid Valerenic acid on Oxidative Stress Induced in HepG2 Cells after Exposure to the Fungicide Benomyl". We are hereby re-submitting a revised manuscript conforming to all of the reviewers' comments. In particular, we have addressed all the reviewers' comments in a point-by-point manner and revisions are indicated in track changes in the revised paper.

We are looking forward to your decision.

Sincerely,

Mehtap Kara, PhD

Istanbul University, Faculty of Pharmacy

Department of Pharmaceutical Toxicology

34116, Beyazit Istanbul - Turkey

Email: mehtap.kara@istanbul.edu.tr

Response to the Reviewers’ comments

The authors would like to thank the Reviewers for their constructive critique to improve the manuscript. We made every effort to address the issues raised and to respond to all comments. The revisions are indicated in track changes in the revised manuscript. Please, find next a detailed, point-by-point response to the reviewers' comments.

Reviewer 1

Comment 1: In this interesting article, the authors investigated in vitro the oxidative stress-related effects of the fungicide benomyl and the possible protective effect of valerenic acid. The research described in this manuscript has been correctly planned and well described.

Response

We would like to thank Reviewer 1 for his/her kind comments.

Comment 2: Were the doses used of VA and benomyl chosen on the basis of previous research? Please write a sentence justifying the doses tested.

Response

The concentrations of benomyl and valerenic acid used in this study were chosen based on the MTT assay results in HepG2 cells. We have added the following sentence in the manuscript, according to the reviewer’s request: «The concentrations of benomyl and VA used in this study were chosen based on the results obtained from the cell viability assay» [2.3. Cell Viability Assay for Benomyl and Valerenic Acid, page 3]. Furthermore, we have added also the following text in order to make clear the experimental procedure for the readership: «For the calculation of the IC50 and IC30 values of benomyl through the MTT assay, solutions with concentrations equal to 2.5, 5, 10, 20, 40 and 60 µM were used, whereas concerning VA, the concentrations of the tested solutions were equal to 300, 400, 500, 600 and 700 µM. After the calculation of the IC50 values of benomyl and VA, further experiments were carried out to determine the IC30 values. The cells were exposed to 6 µM of benomyl (i.e., B6) for 6 h to achieve an induction of cellular stress. Then, in order to evaluate the effects of VA, 100 μM (VA100) and 200 µM (VA200) of VA were chosen for a 24 h-cell exposure following benomyl exposure» [2.3. Cell Viability Assay for Benomyl and Valerenic Acid, page 3].

Comment 3: The discussion is too long. It should be more concise to make it easier to read.

Response

We agree with Reviewer’s comment, therefore we have done our best to shorten the discussion and make it more concise and easier to read.

Comment 4: In the discussion I suggest adding a summary scheme of the results obtained.

Response

We thank Reviewer 1 for this constructive comment. We have added the proposed scheme (i.e., Figure 8) in the manuscript.

Reviewer 2 Report

Paper entitled: ”Ameliorative Effects of the Sesquiterpenoid Valerenic acid on Oxidative Stress Induced in HepG2 Cells after Exposure to the Fungicide Benomyl” (antioxidants-1166645). The subject of the study is investigating the effects of VA against the redox-perturbating role of fungicide benomyl in HepG2 human liver cells as a model. Obtained results seem very interesting however, there are some issues which should be addressed before publication in Antioxidants.

 Please explain, since benovmyl has neurotoxic effects, why this study was not carried out on nerve cell lines.

 2.1

Please provide the cat. No for benomyl – Pestanal and for other tests used

2.3

What was the final concentration of benomyl added to HepG2 cells and how valeric acid was administered to the cells? In what concentration?

Nowhere have I found an explanation of what group B6 means? Please complete

Author Response

Dear editor,

We would like to thank you for your response and for giving us the opportunity to improve and re-submit the paper (antioxidants-1166645) entitled "Ameliorative Effects of the Sesquiterpenoid Valerenic acid on Oxidative Stress Induced in HepG2 Cells after Exposure to the Fungicide Benomyl". We are hereby re-submitting a revised manuscript conforming to all of the reviewers' comments. In particular, we have addressed all the reviewers' comments in a point-by-point manner and revisions are indicated in track changes in the revised paper.

We are looking forward to your decision.

Sincerely,

Mehtap Kara, PhD

Istanbul University, Faculty of Pharmacy

Department of Pharmaceutical Toxicology

34116, Beyazit Istanbul - Turkey

Email: mehtap.kara@istanbul.edu.tr

Response to the Reviewers’ comments

The authors would like to thank the Reviewers for their constructive critique to improve the manuscript. We made every effort to address the issues raised and to respond to all comments. The revisions are indicated in track changes in the revised manuscript. Please, find next a detailed, point-by-point response to the reviewers' comments.

Reviewer 2

Comment 1: The subject of the study is investigating the effects of VA against the redox-perturbating role of fungicide benomyl in HepG2 human liver cells as a model. Obtained results seem very interesting however, there are some issues which should be addressed before publication in Antioxidants.

Response

We would like to thank Reviewer 2 for his/her kind comments. Furthermore, we would like to inform Reviewer 2 that, according to Reviewer’s 1 request, we have added Figure 8 in the manuscript.

Comment 2: Please explain, since benomyl has neurotoxic effects, why this study was not carried out on nerve cell lines.

Response

We thank Reviewer 2 for giving us the opportunity to clarify this matter. This study belongs to a series of studies examining the toxicity mechanisms of benomyl in diverse experimental models and how herbal components could potentially ameliorate its toxic effects. The first study of this series examined, indeed, the toxic roles of benomyl in the SH-SY5Y neuronal cell line (DOI: 10.1016/j.toxrep.2020.04.001). The present study focused on the hepatotoxic effects of benomyl that are of utmost importance, thus the HepG2 was employed. Next studies will also be conducted in neuronal cell lines.

Comment 3: 2.1. Please provide the cat. No for benomyl – Pestanal and for other tests used.

Response

The requested cat No have been added in the manuscript [2.1. Chemicals].

Comment 4: 2.3. What was the final concentration of benomyl added to HepG2 cells and how valeric acid was administered to the cells? In what concentration?

Response

The conditions of exposure of the cells to benomyl and VA are described in the section 2.3. Cell Viability Assay for Benomyl and Valerenic Acid, page 3. However, we have added the following text for extra clarification: «For the calculation of the IC50 and IC30 values of benomyl through the MTT assay, solutions with concentrations equal to 2.5, 5, 10, 20, 40 and 60 µM were used, whereas concerning VA, the concentrations of the tested solutions were equal to 300, 400, 500, 600 and 700 µM. After the calculation of the IC50 values of benomyl and VA, further experiments were carried out to determine the IC30 values. The cells were exposed to 6 µM of benomyl (i.e., B6) for 6 h to achieve an induction of cellular stress. Then, in order to evaluate the effects of VA, 100 μM (VA100) and 200 µM (VA200) of VA were chosen for a 24 h-cell exposure following benomyl exposure» [2.3. Cell Viability Assay for Benomyl and Valerenic Acid, page 3].

Comment 5: Nowhere have I found an explanation of what group B6 means? Please complete.

Response

B6 is referred to the administered concentration of benomyl which is equal to 6 µM. This explanation has been added in the materials and methods section [2.3. Cell Viability Assay for Benomyl and Valerenic Acid, page 3] and in the figure captions.